# Pre-Impressions of the Third COVID-19 Vaccination among Medical Staff: A Text Mining-Based Survey

**DOI:** 10.3390/vaccines10060856

**Published:** 2022-05-26

**Authors:** Yoshiro Mori, Nobuyuki Miyatake, Hiromi Suzuki, Yuka Mori, Setsuo Okada, Kiyotaka Tanimoto

**Affiliations:** 1Department of Hygiene, Faculty of Medicine, Kagawa University, Miki 761-0793, Japan; miyatake.nobuyuki@kagawa-u.ac.jp (N.M.); suzuki.hiromi@kagawa-u.ac.jp (H.S.); 2Sakaide City Hospital, Sakaide 762-8550, Japan; hosp02@city.sakaide.lg.jp (S.O.); taka12ki05@gmail.com (K.T.); 3Graduate School of Health Sciences, Tokushima University, Tokushima 770-8503, Japan; mogeko@hi.enjoy.ne.jp

**Keywords:** text-mining, COVID-19, pregnancy, side effects, third COVID-19 vaccination

## Abstract

The aim of this study was to investigate the pre-impressions of the third Coronavirus disease 2019 (COVID-19) vaccination among Japanese medical staff using quantitative analysis. Among 413 medical staff, 260 (60 men and 200 women) aged 40.4 ± 12.3 years were enrolled in this cross-sectional study. Pre-impressions of the third COVID-19 vaccination were analyzed using the text-mining analysis software, KH coder. Among 260 subjects, 242 (93.1%) agreed to the third vaccination, with the rate being lower among subjects in their 30s (87.3%) than those in the other age groups. The word “side effects” was characteristic of subjects in their 20s and nurses, and “pregnancy” of those in their 30s and administrative staff. Pre-impressions of the third COVID-19 vaccination varied among age groups and different professions. The results obtained provide useful information for promoting the third COVID-19 vaccination to Japanese adults.

## 1. Introduction

Coronavirus disease 2019 (COVID-19) is a global public health issue [1]. The first case of the omicron variant infection was reported in South Africa on 24 November 2021 [2], and by 20 January 2022, 171 countries, including Japan, had reported cases [3,4]. On 1 February 2022, approximately 2.7 million individuals (2.2%) were reportedly diagnosed with COVID-19 infection, including the omicron variant, in Japan [5].

Vaccination is considered to be one of the most effective strategies to prevent COVID-19. Approximately 80% of the Japanese population has been vaccinated twice [6]. However, the need for a third COVID-19 vaccination has increased with the expansion of the omicron variant and has been offered to medical staff since December 2021 [7]. However, the rate of individuals who have received the third vaccination in Japan as of 7 March 2022 was 24.9% (approximately 31.5 million individuals) [8].

We previously reported pre-impressions of the first and second COVID-19 vaccinations among medical staff using a text-mining analysis in Sakaide City Hospital, Sakaide, Japan [9], and main results were as follows: “before the first vaccination, individuals in their 20s concerned about side effects as well as pregnancy. Among medical staff, nurses were more concerned about the side effects of in the first and second vaccinations”. Previous studies reported pre-impressions of COVID-19 vaccinations among medical staff and community-dwelling individuals [10,11,12,13,14,15,16,17,18]. However, apart from our study, pre-impressions of COVID-19 using a text-mining analysis have not yet been investigated. Evaluations of pre-impressions among medical staff may provide useful information for promoting the third vaccination to community-dwelling individuals in Japan in the future.

Therefore, the aim of this study was to investigate the pre-impressions of the third COVID-19 vaccination among medical staff using a text-mining method.

## 2. Materials and Methods

### 2.1. Subjects

Among 413 medical staff at Sakaide City Hospital, Sakaide [19], Japan, 260 (60 men and 200 women, aged 40.4 ± 12.3 years) agreed to answer the survey and provided their written informed consent (Table 1). 

### 2.2. Clinical Parameters 

As previously reported [9], sex, age, and job title were obtained from a self-reported questionnaire. The vaccination rate, those that agreed to be vaccinated, was also calculated by age, sex, and job title. The survey was conducted using the facility’s internal mail system. After stating the purpose of the study in the questionnaire and confirming consent, the respondents were asked to complete the questionnaire by name. The questionnaire could be completed in approximately 5 min or less.

### 2.3. Questionnaire 

Before the third vaccination, an open-ended questionnaire “What do you think of the COVID-19 vaccine?” was surveyed between 2 December and 8 December 2021 (7 days). The same details were used as those described in our previous study [9]. 

### 2.4. Statistical Analysis 

Fisher’s exact test was used to evaluate differences in the percentage of subjects agreeing to the third vaccination stratified by age, sex, and job title, where *p* < 0.05 was significant. Data were analyzed using JMP Pro 15 (SAS Institute Inc., Cary, NC, USA). As previously described [9], a quantitative analysis of pre-impressions toward the third vaccination was evaluated by using text-mining software (KH coder 3.0, Koichi Higuchi, Japan) and the extracted words were translated into English. The software creates a list of words ordered according to their frequencies and interrelationships shown in our previous paper as follows [20,21]: “Step 1: Extract words automatically from data and statistically analyze them to gain a whole picture and investigate the features of the data while avoiding the researcher’s preconceived notions. Step 2: Extract concepts from the data by specifying a coding rule, such as “If a certain expression is found, it is considered to be an occurrence of concept A. Then, the concepts are statistically analyzed to deepen the analysis.” [20]. Correspondence analysis, which is an analytical method that allows the relationship between words to be visualized in a scatter plot, was mainly used in this study. Multivariate data was combined into a new variable, with the variable with the largest contribution on the horizontal axis and the next largest on the vertical axis.

### 2.5. Ethics

Ethical approval was obtained by the Ethical Committee of Sakaide City Hospital, Sakaide, Japan (20 December 2021, approval number: 2021-07).

## 3. Results

The number of subjects (%) analyzed among the total subjects enrolled in the present study is shown in Table 1: 24 medical doctors, 145 nurses, 41 other medical staff, and 50 administrative staff.

The rate of subjects agreeing to the third vaccination is shown in Table 2. The number of subjects who agreed to be vaccinated was 242 (93.1%). The rate of subjects who agreed to be vaccinated was lower among those in their 30s (87.3%) than those in the other age groups. No significant differences were observed in the rate of subjects who agreed to be vaccinated.

Table 3 shows a list of frequently used nouns and adjectival nouns. The total number of words was 6025, with the most frequently used being “vaccination” followed by “adverse reaction”, “side effects”, “anxiety”, and “vaccine”. The word “pregnancy” was the 17th most frequently used word.

In the correspondence analysis (Figure 1 and Figure 2), characteristic words were “side effects” among subjects in their 20s and “pregnancy” among those in their 30s (Figure 1). Regarding job title, characteristic words were as follows: “immunity” and “time” by medical doctors, “side effects” and “scary” by nurses, “vaccination” and “vaccine” by other medical staff, and “fever” and “pregnancy” by administrative staff (Figure 2).

## 4. Discussion

The present study investigated pre-impressions of the third COVID-19 vaccination among medical staff using a text-mining analysis. The word “pregnancy” was characteristic of subjects in their 30s and administrative staff, and “side effects” of subjects in their 20s and nurses.

There are some studies exploring the concerns about COVID-19 vaccination among medical stuff mainly using questionnaire [10,11,12,13]. Spinewine et al. investigated concerns among medical staff about side effects and the impression that the vaccine development was too quick as well as the main motivations for COVID-19 vaccination [10]. Gravlee et al. found that the most identified barrier to COVID-19 vaccination among pharmacists was patient willingness [11]. Among healthcare professionals (physicians, dentists, and pharmacists), age 45 years or older, no fear of vaccine safety, and information received from public health authorities were factors associated with COVID-19 vaccination acceptance [12]. Adequate safety information and knowledge of COVID-19 vaccination were important factors contributing to hesitancy among health care providers [13]. In this study, it is noteworthy that we first investigated pre-impressions by using a text-mining procedure of the third COVID-19 vaccination.

A previous study reported that younger generations (18–40 years old) and a lower education level were characteristic of individuals who were hesitant to receive the first and second vaccinations [14]. The rates of vaccination for the COVID-19 booster, influenza, and combination influenza-COVID-19 booster vaccines were previously shown to be lower among female, Black/African American, Native American/American Indian, and rural respondents [15]. The side effects of the first and second COVID-19 vaccinations, uncertainty regarding safety, and the belief that the COVID-19 vaccination is not necessary were identified as the main reasons for not receiving the third vaccination [16]. Sugawara et al. found that medical students’ willingness to receive a third COVID-19 vaccination in Japan was significantly related to their grade, positive attitude toward vaccination, belief in the preventive effect of COVID-19 vaccination, and concern about the rapid development of the COVID-19 vaccine and immune persistence [17]. Another main reason for refusing to receive the booster vaccination was the belief that the previous vaccination provides sufficient protection [18].

In the present study, the rate of subjects who agreed to be vaccinated was 93.1% and was lower among those in their 30s than in other age groups. Furthermore, subjects in their 30s and administrative staff expressed concerns about pregnancy. The term “side effects” was fundamentally characteristic of subjects in their 20s and nurses, which is consistent with our previous findings [9]. The word “pregnancy” was also characteristic of subjects in their 20s and nurses receiving the first and second vaccinations [9]. The word “pregnancy” shifted from subjects in their 20s to those in their 30s and from nurses to administrative staff. Among medical staff, administrative staff were considered to be similar to community-dwelling individuals. Therefore, the results obtained from the present study provide useful information for promoting the third vaccination to the general population in Japan. Regarding the human papillomavirus vaccination for girls, the vaccination rate was increased by education and the transmission of information [22]. Furthermore, the rate of seasonal influenza vaccination was increased among pregnant women by education [23]. Therefore, the provision of more intensive information on the third COVID-19 vaccination, including pregnancy to individuals in their 30s and side effects to those in their 20s, is important.

There are some limitations that need to be addressed. This was a single institutional cross-sectional study. Furthermore, the subjects enrolled were medical staff with more medical knowledge than community-dwelling individuals. Therefore, the enrolled subjects in this were thought to be more health-conscious than the average person. Moreover, the survey period was only seven days by using the facility’s internal mail system with open-ended questionnaire, and the response rate to the questionnaire was low (63%). Nevertheless, the results including administrative staff, who were thought to be comparatively similar to community-dwelling people, obtained provide useful information for promoting the third COVID-19 vaccination throughout Japan.

In summary, we evaluated pre-impressions of the third COVID-19 vaccination and found that the word “pregnancy” was characteristic of individuals in their 30s and administrative staff, while “side effects” was characteristic of those in their 20s and nurses.

## 5. Conclusions

By analyzing the questionnaire responses with KH Coder, pre-impressions of the third COVID-19 vaccination varied among age groups and different professions. The results obtained provide useful information for promoting the third COVID-19 vaccination to Japanese adults, especially in women in their 30s and younger generations.

## Figures and Tables

**Figure 1 vaccines-10-00856-f001:**
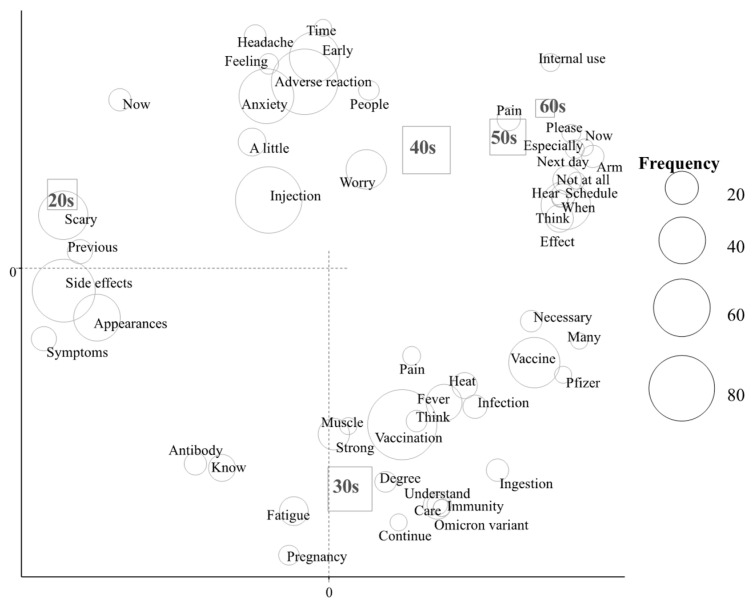
Correspondence analysis of the relationship between frequently used words and age.

**Figure 2 vaccines-10-00856-f002:**
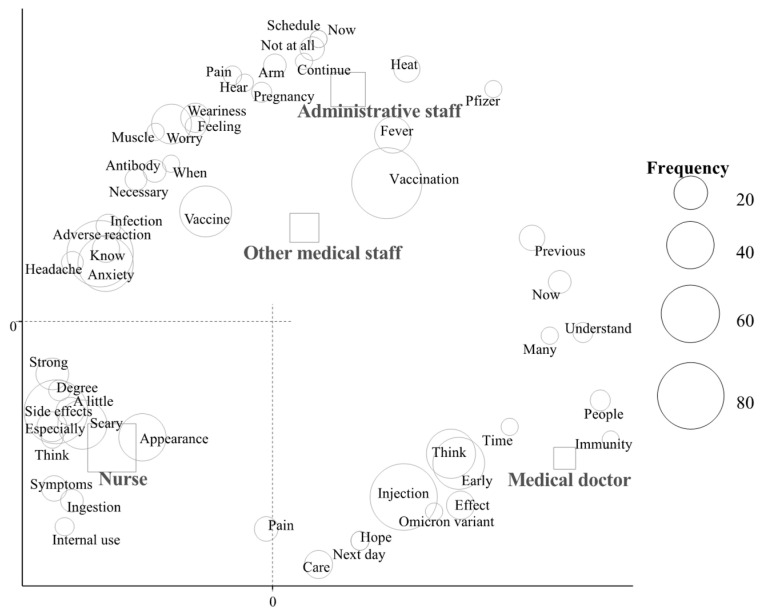
Correspondence analysis of the relationship between frequently used words and job title.

**Table 1 vaccines-10-00856-t001:** The subjects enrolled in this study.

	Total		The Analyzed Subjects	
	(413 Subjects)		(260 Subjects)	
	Number of Subjects	%	Number of Subjects	%
**Age**				
20–29	103	23.5	64	24.6
30–39	90	21.4	63	24.2
40–49	122	31.0	75	28.8
50–59	66	15.8	39	15.0
60–	32	8.3	19	7.3
**Sex**				
Men	104	25.8	60	23.1
Women	309	74.2	200	76.9
**Job title**				
Medical doctor	48	13.2	24	9.2
Nurse	220	50.9	145	55.8
Other medical staffs	51	12.4	41	15.8
Administrative staffs	94	23.5	50	19.2

**Table 2 vaccines-10-00856-t002:** Subjects who have agreed to be vaccinated.

	Number of Subjects	%	*p*
**Age**			
20–29	61	95.3	0.236
30–39	55	87.3	
40–49	69	92.0	
50–59	38	97.4	
60–	19	100.0	
**Sex**			
Men	59	98.3	0.083
Women	183	91.5	
**Job title**			
Medical doctor	24	100.0	0.208
Nurse	134	92.4	
Other medical staffs	40	97.6	
Administrative staffs	44	88.0	

Comparison was used by Fisher’s exact test. %: Percentage of those willing to be vaccinated among those who agreed to the study.

**Table 3 vaccines-10-00856-t003:** List of frequently used words among nouns and adjectival nouns.

	Word	Word Counts
The total number of words		6025
1st	Vaccination	89
2nd	Adverse reaction	80
3rd	Side effects	73
4th	Anxiety	56
5th	Vaccine	47
6th	Worry	29
7th	Fever	24
8th	Fatigue	15
9th	Effect	14
10th	Symptoms	11
10th	Previous	11

## Data Availability

Not applicable.

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
