# Peer review of "Pre-Impressions of the Third COVID-19 Vaccination among Medical Staff: A Text Mining-Based Survey"

_vaccines, 2022, doi:10.3390/vaccines10060856_

Round 1

Reviewer 1 Report

The frequency plots are a little confusing.  Not clear what the axises are and how they are generated.

Author Response

Thank you for the thoughtful and constructive feedback you provided regarding our manuscript, Pre-impressions of the Third COVID-19 Vaccination among Medical Staff: A Text Mining-based Survey (vaccines-1708999)

Our responses to the referees’ comments are as follow:

==================================

Reviewer 1

  • The frequency plots are a little confusing.  Not clear what the axises are and how they are generated.

According to your advice, we have enlarged the text in the figure and corrected overlapping text. Further details about the correspondence analysis were added to the method as follows:

Multivariate data was combined into a new variable, with the variable with the largest contribution on the horizontal axis and the next largest on the vertical axis.

Again, thank you for giving us the opportunity to strengthen our manuscript with your valuable comments and queries. We have worked hard to incorporate your feedback and hope that these revisions persuade you to accept our submission.

Reviewer 2 Report

  • Even if the methods was already explained in a previous article, for a better understanding of the study I suggest including more detail of this study in the Materials and Methods section.
  • The authors should add more information in the Manuscript about the sampling strategy.
  • The authors should add more information about the survey instrument. Is the Questionnaire anonymous? what was the time for completion, how was it withdrawn by the researchers?
  • Please check the total percentages in the tables because it does not always reach 100%.
  • Are there any differences between respondents and non-respondents? if possible, include the Results in the Manuscript.
  • The Discussion is very poor and should be expanded with national and international references.
  • The authors should address more thoroughly in the Discussion section the limits of this study regarding the response rate, the representativeness of the sample, and the methods to collect data.
  • The authors should explain why the results of this study on medical staff could be useful for promoting COVID-19 vaccination among the general adult population.

Author Response

Thank you for the thoughtful and constructive feedback you provided regarding our manuscript, Pre-impressions of the Third COVID-19 Vaccination among Medical Staff: A Text Mining-based Survey (vaccines-1708999)

Our responses to the referees’ comments are as follow:

Reviewer 2

  • Even if the methods was already explained in a previous article, for a better understanding of the study I suggest including more detail of this study in the Materials and Methods section.

According to your advice, we have described this study in detail in the Materials and Methods section.

  • The authors should add more information in the Manuscript about the sampling strategy.

According to your advice, we have described this study in detail in the Materials and Methods section.

  • The authors should add more information about the survey instrument. Is the Questionnaire anonymous? what was the time for completion, how was it withdrawn by the researchers?

According to your advice, we stated as follows in Clinical Parameters: The survey was conducted using the facility's internal mail system. After stating the purpose of the study in the questionnaire and confirming consent, the respondents were asked to complete the questionnaire by name. The questionnaire could be completed in approximately 5 minutes or less.

  • Please check the total percentages in the tables because it does not always reach 100%.

We have included the following in Table 2 for lack of explanation: Percentage of those willing to be vaccinated among those who agreed to the study

  • Are there any differences between respondents and non-respondents? if possible, include the Results in the Manuscript.

Only those who consented to the survey were analyzed, so the results of those who did not consent are not available. Therefore, they cannot be included in the results, etc.

  • The Discussion is very poor and should be expanded with national and international references.

According to your advice, we added the domestic and international literatures in the discussion.

  • The authors should address more thoroughly in the Discussion section the limits of this study regarding the response rate, the representativeness of the sample, and the methods to collect data.

Following your advice, we have described in more detail at the marginal points of the study in the discussion as follows:

There are some limitations that need to be addressed. This was a single institutional cross-sectional study. Furthermore, the subjects enrolled were medical staff with more medical knowledge than community-dwelling individuals. Therefore, the enrolled subjects in this were thought to be more health-conscious than the average person. Moreover, the survey period was only seven days by using facility's internal mail system with open-ended questionnaire, the response rate to the questionnaire was low (63%).

  • The authors should explain why the results of this study on medical staff could be useful for promoting COVID-19 vaccination among the general adult population.

Following your advice, we have added to the discussion as follows:

Nevertheless, the results including administrative staff, who were thought to be comparatively similar to community-dwelling people, obtained provide useful information for promoting the third COVID-19 vaccination throughout Japan.

Again, thank you for giving us the opportunity to strengthen our manuscript with your valuable comments and queries. We have worked hard to incorporate your feedback and hope that these revisions persuade you to accept our submission.

Reviewer 3 Report

Type: Communication

Title: Pre-impressions of the Third COVID-19 Vaccination among Medical Staff: A Text Mining-based Survey

Abstract

The present study examined pre-impressions of the third Coronavirus disease 2019 (COVID-19) vaccination among Japanese medical staff using a text mining-based analysis. Among 413 medical staff, 260 (60 men and 200 women) aged 40.4±12.3 years were enrolled in this cross-sectional study. Pre-impressions of the third COVID-19 vaccination were analyzed using the text-mining analysis software, KH coder. Among 260 subjects, 242 (93.1%) agreed to the third vaccination, with the rate being lower among subjects in their 30s (87.3%) than those in the other age groups. The word “side effects” was characteristic of subjects in their 20s and nurses, and “pregnancy” of those in their 30s and administrative staff. Pre-impressions of the third COVID-19 vaccination varied among age groups and different professions. The results obtained provide useful information for promoting the third COVID-19 vaccination to Japanese adults.

This work has the same method whom the author team published recently:

https://www.mdpi.com/2076-393X/9/11/1282/htm

A Text Mining-Based Survey of Pre-Impressions of Medical Staff toward COVID-19 Vaccination in a Designated Medical Institution for Class II Infectious Diseases

Similar headings, as per the previous papers were used:

  1. Materials and Methods

2.1. Subjects

2.2. Clinical Parameters

2.3. Questionnaire

2.4. Statistical Analysis

2.4. Ethics

I would like to highlight the current state of statistical analysis, as shown below, is not complete:

Fisher’s exact test was used to evaluate differences in the percentage of subjects 64 agreeing to the third vaccination stratified by age, sex, and job title, where p<0.05 was 65 significant. Data were analyzed using JMP Pro 15 (SAS Institute Inc., Cary, NC, USA). The 66 self-reported questionnaire was analyzed using text-mining analysis software (KH coder 67 3.0, Kouichi Higuchi, Japan [16,17]), as previously described [9]. A list of words ordered 68 by their frequencies and a correspondence analysis were used in the present study [11].( https://www.mdpi.com/2076-393X/9/11/1282/htm)

A more complete description, as per authors’ paper, should be used:

Data were expressed as the mean ± standard deviation (SD). The χ2 test was used to evaluate differences in vaccination rates among groups based on sex, age and job title, where p < 0.05 was considered to be significant. A quantitative analysis of impressions toward the vaccination was performed using text mining software (KH coder 3.0, Koichi Higuchi, Japan) and extracted words were converted into English. KH Coder produces a list of words ordered according to their frequencies and interrelationships as previously described as follows [22,23]: “Step 1: Extract words automatically from data and statistically analyze them to obtain a whole picture and explore the features of the data while avoiding the prejudices of the researcher. Step 2: Specify coding rules, such as “if there is a particular expression, we regard it as an appearance of the concept A”, and extract concepts from the data. Then, statistically analyze the concepts to deepen the analysis” [22]. We used a correspondence analysis, which is an analytical method that allows the relationship between words to be visualized in a scatter plot.

  1. Conclusions

By analyzing the questionnaire responses with KH Coder, pre-impressions of the 142 third COVID-19 vaccination varied among age groups and different professions. The re-143 sults obtained provide useful information for promoting the third COVID-19 vaccination 144 to Japanese adults

Comment: the conclusion is rather generic. This is especially true when in juxtaposition of the previous paper:

  1. Conclusions

By analyzing the questionnaire responses with KH Coder, we were able to extract the concerns of healthcare workers before vaccination. These data may be useful for promoting COVID-19 vaccination to the public, particularly the younger generation and women. (https://www.mdpi.com/2076-393X/9/11/1282/htm)

Author Response

Thank you for the thoughtful and constructive feedback you provided regarding our manuscript, Pre-impressions of the Third COVID-19 Vaccination among Medical Staff: A Text Mining-based Survey (vaccines-1708999)

Our responses to the referees’ comments are as follow:

==================================

Reviewer 3

  • I would like to highlight the current state of statistical analysis, as shown below, is not complete:

Fisher’s exact test was used to evaluate differences in the percentage of subjects 64 agreeing to the third vaccination stratified by age, sex, and job title, where p<0.05 was 65 significant. Data were analyzed using JMP Pro 15 (SAS Institute Inc., Cary, NC, USA). The 66 self-reported questionnaire was analyzed using text-mining analysis software (KH coder 67 3.0, Kouichi Higuchi, Japan [16,17]), as previously described [9]. A list of words ordered 68 by their frequencies and a correspondence analysis were used in the present study [11].( https://www.mdpi.com/2076-393X/9/11/1282/htm)

A more complete description, as per authors’ paper, should be used:

Data were expressed as the mean ± standard deviation (SD). The χ2 test was used to evaluate differences in vaccination rates among groups based on sex, age and job title, where p < 0.05 was considered to be significant. A quantitative analysis of impressions toward the vaccination was performed using text mining software (KH coder 3.0, Koichi Higuchi, Japan) and extracted words were converted into English. KH Coder produces a list of words ordered according to their frequencies and interrelationships as previously described as follows [22,23]: “Step 1: Extract words automatically from data and statistically analyze them to obtain a whole picture and explore the features of the data while avoiding the prejudices of the researcher. Step 2: Specify coding rules, such as “if there is a particular expression, we regard it as an appearance of the concept A”, and extract concepts from the data. Then, statistically analyze the concepts to deepen the analysis” [22]. We used a correspondence analysis, which is an analytical method that allows the relationship between words to be visualized in a scatter plot.

According to your advice, we have added to the Statistical Analysis in Materials and Methods as follows:

Fisher’s exact test was used to evaluate differences in the percentage of subjects agreeing to the third vaccination stratified by age, sex, and job title, where p<0.05 was significant. Data were analyzed using JMP Pro 15 (SAS Institute Inc., Cary, NC, USA). As previously described [9], a quantitative analysis of pre-impressions toward the third vaccination was evaluated by using text mining software (KH coder 3.0, Koichi Higuchi, Japan) and extracted words were translated into English. The software creates a list of words ordered according to their frequencies and interrelationships shown in our previous paper as follows [20, 21]: “Step 1: Extract words automatically from data and statistically analyze them to gain a whole picture and investigate the features of the data while avoiding the researcher's preconceived notions. Step 2: Extract concepts from the data by specifying a coding rule, such as "If a certain expression is found, it is considered to be an occurrence of concept A. Then, the concepts are statistically analyzed to deepen the analysis.” [20]. Correspondence analysis, which is an analytical method that allows the relationship between words to be visualized in a scatter plot, was mainly used in this study. Multivariate data was combined into a new variable, with the variable with the largest contribution on the horizontal axis and the next largest on the vertical axis

Conclusions

  • Comment: the conclusion is rather generic. This is especially true when in juxtaposition of the previous paper:

Following your advice, we have added to the Conclusions as follows:

By analyzing the questionnaire responses with KH Coder, pre-impressions of the third COVID-19 vaccination varied among age groups and different professions. The results obtained provide useful information for promoting the third COVID-19 vaccination to Japanese adults, especially in women in her 30’s and younger generation.

Again, thank you for giving us the opportunity to strengthen our manuscript with your valuable comments and queries. We have worked hard to incorporate your feedback and hope that these revisions persuade you to accept our submission.

Round 2

Reviewer 2 Report

None

Reviewer 3 Report

All comments have been addressed